# Experiences of Impacted Foetal Head: Findings from a Pragmatic Focus Group Study of Mothers and Midwives

**DOI:** 10.3390/ijerph20217009

**Published:** 2023-11-02

**Authors:** Annette L. Briley, Sergio A. Silverio, Andrew H. Shennan, Graham Tydeman

**Affiliations:** 1Caring Futures Institute, College of Nursing and Health Sciences, Flinders University, Adelaide, SA 5042, Australia; annette.briley@flinders.edu.au; 2Department of Women & Children’s Health, School of Life Course & Population Sciences, King’s College London, London SE1 1UL, UK; andrew.shennan@kcl.ac.uk; 3School of Psychology, Faculty of Health, Liverpool John Moores University, Liverpool L3 3AF, UK; 4Maternity Services, Victoria Hospital, NHS Fife, Kirkcaldy KY2 5AH, UK; grahamtydeman@gmail.com

**Keywords:** maternity care, midwives, labour, birth, impacted foetal head, complications of labour, complications in birth, qualitative research

## Abstract

Introduction: We aimed to explore the lived experiences of caesarean birth complicated by impaction of the foetal head, for mothers and midwives. Methods: A pragmatic, qualitative, focus group study of mixed-participants was conducted, face-to-face. They were postpartum women (n = 4), midwives (n = 4), and a postpartum midwife (n = 1) who had experience of either providing care for impacted foetal head, and/or had experienced it during their own labour, in Fife, United Kingdom. Data were transcribed and were analysed using template analysis. Results: Three main themes emerged through analysis: (i) current knowledge of impacted foetal head; (ii) current management of impacted foetal head; and (iii) experiences and outcomes of impacted foetal head. Each theme was made up of various initial codes when data were analysed inductively. Finally, each theme could be overlaid onto the three core principles of the Tydeman Tube: (1) to improve outcomes for mother and baby in the second stage of labour; (2) to reduce the risk of trauma to mother and baby in complicated births; and (3) to increase respectful care for women in labour; thus allowing for a neat analytic template. Conclusion: A lack of consensus regarding definition, management, and training were highlighted by the midwives. Women anticipated caesarean birth in late labour as straightforward and were therefore unaware of this potential complication. Women and midwives would welcome any new device to facilitate delivery of the impacted foetal head (IFH) as long as it is fully evaluated prior to widespread introduction. Women were not averse to being part of this evaluation process.

## 1. Introduction

In the UK, more than 25% of babies are born by caesarean section (CS) [1], increasingly, these occur in the late first or second stage of labour, when the foetal head is deep within the maternal pelvis. It has been estimated that full-dilatation caesarean sections (FDCS) are undertaken in 2–5% of cases [2,3,4,5,6] and in 15% of all intrapartum caesareans [7].

Caesarean section in late labour has long been reported as technically more challenging for the obstetrician, but also associated with increased maternal and neonatal damage [8]. Reported maternal complications include excessive blood loss, extensions to incisions and infections; neonatal complications include damage to the head and face, sub-gluteal haemorrhage, nerve damage and in rare cases enucleation and rarely foetal de-mise [9]. Difficulty in delivering the head, or impacted foetal head (IFH), has been reported as occurring in 9% and 11.3%, respectively, of all emergency caesarean births [5,10]. Yet, unlike shoulder dystocia, with a similar incidence (12%), there are no established national protocols or training regarding this intrapartum complication [10].

Similarly, there is no consensus on optimal management of delivering the foetal head when it is low in the maternal pelvis. One recent UK survey concluded that the ‘push technique’, whereby an assistant inserts a hand to push the baby up vaginally, whilst the surgeon attempts delivery through the abdominal incision, was commonly used by UK obstetricians and midwives (84% versus 69%, respectively) [11]. Whilst considered a complication of caesarean section at full dilatation, there is increasing evidence of difficulty delivering the foetal head at lesser cervical dilatations [6,7].

The reasons for the increasing incidence of this clinical problem are complex and multifactorial. Maternal BMI and foetal macrosomia are not contributing factors. It has been suggested that changes in specialist training, where CS is favoured over instrumental vaginal birth, is contributary [12] and the use of syntocinon to induce or augment labour has also been cited [10].

Using bespoke online surveys, designed by a multi-disciplinary team including consumers, Hanley et al. [11] concluded that most doctors and midwives who responded emphasised the divergence of techniques used. The majority of parents who responded stated that whichever technique of device the doctor was most familiar with was the most acceptable option [11].

There is one commercially available device to aid delivery of the IFH: The Fetal Pillow^®^ [13]. The Fetal Pillow is a balloon device, inserted vaginally prior to uterine incision to aid elevation of the foetal head [13]. This device is inserted into the vagina of the woman immediately prior to CS where impaction is suspected and is filled with 180 mL of saline, which is hoped to push the head up out of the pelvis sufficiently to make delivery easier. Although results have been variably reported [14,15,16] and its use is often limited by resource availability, the Manufacturer’s Brochure claims it reduces associated maternal morbidities, hospital stay and resource use. Despite no evidence, it also claims to reduce neonatal morbidities [13].

A novel device—The Tydeman Tube—is currently in progress, due to be commercially available by the end of 2023, and is an alternate device with similar aims to The Fetal Pillow, but the elevation is achieved by an assistant. It is a semi-rigid silicone tube with a cup on the end that makes contact with the foetal head. It is also inserted in the vagina just prior to CS, but the amount and direction of elevation is achieved by an assistant ‘pushing up’. Unlike the pillow, it can be used in cases where there is less than full dilation and also, being hollow, prevents any suction effect which can in some cases hinder delivery. The Tydeman Tube aims to assist elevation by enabling a ‘push-up’ from below by insertion of tube into the vagina, and the flanged head of the device disperses pressure ameliorating potential damage to the foetal head. The tube enables air to enter and break the vacuum that can form between maternal tissues and the foetal head. This surgical-quality silicone tube can be inserted prior to incision, and therefore when difficulty is anticipated, or when it is encountered during the caesarean section.

The aim of the current study was to hear the lived experience of a caesarean birth complicated by IFH from midwives’ and mothers’ perspectives. Adopting a widely utilised qualitative research methodology, we sought to better understand the experiences of both women who have been told during a recent labour that there had been some difficulty delivering their baby’s head, and the midwives who have had to provide clinical care for impacted foetal heads in practice. Our findings contribute to ongoing development of clinical practice and novel clinical tools and mechanisms in order to provide optimum, less invasive, and respectful care for women who experience difficulties during labour and birth.

## 2. Materials and Methods

### 2.1. Design

This work was carried out as both practically (research approach) and theoretically (theoretical perspective) pragmatic. The research team’s philosophical underpinning was ontologically and epistemologically pragmatic; and in terms of positionality, the researchers were ‘absent’ in their positional relationship to the data (i.e., we did not experience the phenomenon of interest), and reflexively ordered in judgement (according to the social and policy norms of complex health needs). Together, this rendered our research paradigmatically pragmatic [17].

We, therefore, took a pragmatic decision to run a mixed focus group representing a ‘newly formed group’ [18], comprising both postpartum women and midwives (which included a midwife who had also recently given birth). The focus group facilitated shared discussion and understanding of the experience of receiving or delivering information, suggesting there had been difficulty delivering the baby’s head during labour, and the subsequent care that was received or provided to rectify the complication of an impacted foetal head. The focus group followed a structured topic guide which asked participants questions about experiences of difficulty delivering the foetal head; perceptions of women’s partners’ feelings during the birth complication; their knowledge of foetal pillows; and the potential for a clinical trial using the Tydeman Tube.

### 2.2. Ethical Considerations

Ethical approval was sought from the East of Scotland Research Ethics Service [ref:-LR/AG/14/GA/0029], whose Scientific Officer and Senior Co-Ordinator reviewed the request and advised that it did not require National Health Services (NHS) ethical review under the terms of the Governance Arrangements for Research Ethics Committees, and was subsequently approved by NHS Fife Trust’s Research and Development Department, as a clinical audit. Oral consent was taken from participants at the beginning of the focus group.

### 2.3. Setting

Fife is an area in South East Scotland with a population of approximately 375,000. It is largely rural, with the majority of the population concentrated in the towns of Kirkcaldy, Dunfermline, and Glenrothes. In common with other areas of Scotland, the unemployment rate is 9.4% and income deprivation is 11.9%, and 20% of residents live in the most deprived areas [19]. In 2020, 3,143 babies were born in Fife, a 5.5% decrease from 2019 [20].

### 2.4. Participants and Recruitment

In keeping with our pragmatic research approach, recruitment took place in one hospital Trust, and was undertaken using a homogenous purposive sampling approach, which aims “*to select a group of cases with similar backgrounds and experiences*” in order to allow for group interviewing [21] (p. 70). In our case, it consisted of a narrow set of questions around the specific experience of having or providing care for an impacted foetal head during labour and birth, which will be used to inform future clinical practice [22,23].

Women were eligible if they experienced Caesarean section during labour and particularly where difficulty delivering the foetal head was documented in their medical records (electronic or paper). Midwives were eligible if they had any experience with delivering an impacted foetal head and agreed to be part of the focus group.

All participants were white (n = 9). One mother was also a midwife. All mothers (n = 5) were primiparous, and all babies were singletons, born by emergency Caesarean section in established labour, one after a failed forceps delivery. Cervical dilatation at Caesarean section ranged from 6cm to fully dilated. All midwives (n = 5) worked at the same Hospital Trust, with two from the labour ward, one from the midwifery-led unit, and the other working as a rotational midwife. All midwives bar one reported more than five years’ experience of working as a midwife at the time of the focus group (ranging from 4–32 years; Mean = 13 years; Median = 10 years) Four of the midwives had Bachelor’s degrees, and two had dual registration. The midwife, who was also a mother, with experience of impacted foetal head had 10 years clinical experience and had worked in all areas but had been working on labour ward prior to maternity leave.

### 2.5. Data Collection

Potential participants were e-mailed by the research team and were sent an electronic questionnaire and participant information leaflet, which were completed prior to the focus group. The focus group was held in-person, and was facilitated by two members of the research team [A.L.B., G.T.]—one who worked at the Hospital as an obstetrician [G.T.], the other a midwife from a collaborating institution [A.L.B.]. Discussion points included how women had felt when they were told there had been some difficulty in delivering the head; whether women had been aware of any techniques or devices that were used to help deliver the baby’s head; how their partners had felt during the operation; whether any of those present has ever heard of or used a Fetal Pillow (and asked if they thought such a device might be a good idea and asked for their opinions of it); and whether both groups would take part in a clinical trial to evaluate the Tydeman Tube. The focus group lasted approximately 90 min and was audio-recorded. Following our pragmatic research approach, the audio was transcribed [A.L.B.] using selective (or ‘edited’) transcription, which always “*corrects for neologisms or incorrectly used and portmanteau words, whilst omitting all contextual matter.*” [24] (p. 44).

### 2.6. Data Analysis

Template analysis [25] was chosen as the most appropriate and pragmatic analytic methodology with which to evaluate experiences of IFH, in relation to the here core principles for which the Tydeman Tube had been developed: (1) to improve outcomes for mother and baby in the second stage of labour; (2) to increase respectful care for women in labour; and (3) to reduce the risk of trauma to mother and baby in complicated births.

As a method with which to analyse qualitative data, template analysis has philosophical plasticity [26], and in this study, was employed with a pragmatic perspective [17], whereby our straightforward questioning and subsequent analysis “*grapples with analyzing contemporary social issues*”—in our case, a health issue related to complicated birth—“*and engages with themes of social inequality, power, and politics*” [27] (p. 11). Template analysis follows a methodical procedure, which in brief consists of the following steps: (re)familiarisation with the data; a preliminary coding of the data; thematic organisation; defining a preliminary coding template; application of this preliminary template; finalising the template; and applying the template to the full dataset [28]. A key feature of template analysis is that the preliminary template, which is devised, is then modified and augmented to ensure analytic completeness [25].

Our use of template analysis with these data proceeded as follows. The analyst who had conducted the interviews [A.L.B.] checked the transcript for accuracy against the audio recording, making allowances for missing contextual matter due to the selective transcription format. This also allowed for re-familiarisation with the data. The analyst who had not conducted the interviews [S.A.S.] also read the checked transcript to familiarise himself with the data. Data were analysed manually using Microsoft Word [A.L.B., S.A.S.], which facilitated a pragmatic and rapid analysis and the evaluation of the data [29]. Research utilising template analysis maintains rigour by iteratively engaging with data and repeating the analytical processes throughout all analytical steps. Brooks et al. [28], also suggest that iterative coding permits the more thorough and systematic analysis of data, whilst encouraging accuracy, checking in later stages to provide certainty and confidence in assessed thematic saturation (i.e., where the final themes are supported robustly by data). Following instruction for accurate template analysis methodology [26], both the coding and extraction of quotations for presentation in the final write-up were conducted iteratively [A.L.B., S.A.S.], involving reading and re-reading data, and making analytic and interpretive comparisons.

## 3. Results

The preliminary coding template was tested [A.L.B.], and modified [S.A.S., A.L.B.], before being re-applied to the data [A.L.B.]. The preliminary coding template included nine specific thematic concepts which—upon modification of the analytic template—were then grouped in order to map onto the key principles for which the Tydeman Tube had been developed (see Table 1). The three themes in the final template were (1) current knowledge of impacted foetal head; (2) current management of impacted foetal head; (3) experiences and outcomes of impacted foetal head. The most eloquent and illustrative quotations from the data were chosen for reporting (see Table 1), and are presented with corresponding participant identifiers.

### 3.1. Current Knowledge of Impacted Foetal Head

The first thematic area within the template addressed both women’s and staff’s knowledge of impacted foetal heads during labour. Preliminary analysis of the data identified manoeuvres employed in order to safely deliver an impacted foetal head to be an important factor of this part of the template, as well as knowledge of the devices used to assist the delivery of the foetal head. Finally, patient perspectives of complicated birth were also captured under this part of the template, demonstrating patient-level knowledge of the potential complication of labour. Together, these three areas could be grouped under the first theme of the template (current knowledge of impacted foetal head) and contribute to the first core principle of the Tydeman Tube: To improve outcomes for mother and baby in the second stage of labour.

### 3.2. Current Management of Impacted Foetal Head

Within our template analysis, the second thematic area concerned more perceptive and experiential data from mothers and midwives. Our initial coding pass identified both staff- and patient-level expectations of the associated trouble there could be in delivering the foetal head, as well as experiences—either during their own labour, or assisting a woman’s labour—of an impacted foetal head. Additionally, this area of the template addressed perceptions of a midwife’s role in the management of an impacted foetal head. Taken together, these three areas were grouped into the second theme of the template (current management of impacted foetal head) and contribute to the second core principle of the Tydeman Tube: to reduce the risk of trauma to mother and baby in complicated births.

### 3.3. Experiences and Outcomes of Impacted Foetal Head

Finally, the last thematic area of the template analysis encompassed external factors and perceptions of experience. Our first code of the data allowed us to group midwives’ and mothers’ perspectives of the experience of ‘pushing up’ and on debriefing around an impacted foetal head and difficult labour and birth. In addition, this was the part of the template analysis which also identified partners’ perceptions of difficult labour and complex birth. When grouped together, these areas formed the third and final theme of the template (experiences and outcomes of impacted foetal head) and contributed to the third core principle of the Tydeman Tube: to increase respectful care for women in labour.

## 4. Discussion

### 4.1. Main Findings

These data demonstrate that women’s knowledge of IFH at CS is limited. Most women and their partners had considered that a caesarean section in late labour would be straightforward. There has been much controversy regarding information sharing about adverse events that might occur during pregnancy and labour, with clinicians blamed for turning the antenatal period from a time of excited expectation to one of anxiety for parents-to-be.

The UK National Institute for Health and Clinical Excellence (NICE) acknowledges that women who develop intrapartum complications may be less able to assimilate information, due to anxiety, advocating healthcare professionals adapt communication techniques to ensure women are aware of risks and benefits and can make informed decisions [30]. Given the unpredictability of difficulty delivering the foetal head at late caesarean section, and the timing of this complication, it is important that women are informed and engaged in the process, but the timing and detail of such information remains contentious.

Some women described a lack of understanding regarding what was happening, which raises further concerns around informed consent, but all those interviewed appreciated the calm voices of doctors and midwives informing, reassuring, and supporting them throughout the process.

The impact on partners witnessing the event needs consideration: although no partners were present for the interviews, all the women mentioned they had found the situation difficult. Given growing evidence of heightened paternal distress when unexpected complications occur [31,32,33] and the acknowledged precursors of paternal childbirth trauma identified as lack of support from health care professionals, limited involvement in the birth and no opportunities to talk about the events afterwards [34], it is important to consider the impact on those witnessing such events. Several authors have suggested the impact for men of witnessing complicated intrapartum events can lead to short- and longer-term changes in their mental health, and adversely impact their relationships with their partner and child [35,36]. Historically, incidence of paternal childbirth-related post-traumatic stress symptoms, assessed using questionnaires, has been estimated as 0–8% [37], but more recently using the validated City BiTS (Partner version) tool and excluding those whose symptoms could have other causes, 22% fathers reported symptoms that caused them distress [36].

The midwives in the focus group had all witnessed or assisted in caesarean births in which they had encountered difficulty delivering the foetal head. All had been asked, at least once, to assist by providing a ‘push up’. None had received training about this, or any other, manoeuvre. One midwife had seen a Foetal Pillow used in a previous hospital, but was not sure how it worked. None felt confident in this situation, regardless of the duration of providing intrapartum care. This study concurs with others regarding the incidence of this complication, lack of consensus regarding management and absence of training for this emergency [5,6,7,11,38].

All the midwives in the focus group felt training and protocols around the management of the IFH should be part of undergraduate and ongoing mandatory education. Several agreed this should be multi-disciplinary. The mothers interviewed were surprised by the lack of recognised procedures, but all felt their case was managed well and the team worked expertly together. Some did, however, mention the non-verbal cues emitted by staff, which indicated concern or anxiety. It is important that staff are cognisant of these involuntary signals which women and their partners pick up on [33]. It is possible that these would be reduced with the confidence developed from unified strategies and training to deal with this complication.

Experience regarding use of the commercially available Foetal Pillow^®^ was limited, due, in part, to it not being provided within the Trust. Several respondents knew of its existence, although opinion regarding its efficacy varied according to what they had read or heard. This concurs with the available contradictory evidence regarding its effectiveness [5,14,39]. Given recent evidence that IFH occurs prior to full dilatation [5,11] the Foetal Pillow may not be appropriate for use in all cases, as it can only be used at full cervical dilatation. It also needs insertion prior to the onset of CS; therefore, impaction needs to be anticipated and, although there are predisposing factors, many of these cases are not predictable. In response to clinical incidents and litigation, some hospitals are advocating the use of the Foetal Pillow at all caesareans in late labour, but this increases the cost of all caesareans, when evidence shows the device may only be required for one in five [7].

Acceptability for a new device was high. The midwives liked the idea that assistance could be provided without a difficult vaginal examination, and where there was less likelihood of exerting too much, or too little, pressure on the foetal head. The women also agreed that something that was less invasive than a digital manoeuvre was preferable, although there was consensus that, by this time, they just wanted their baby to be born, and felt whatever their doctor was most comfortable with was acceptable to them.

The explanation of how the Tydeman Tube works and its design were also acceptable to midwives and mothers. All respondents emphasised the importance of any new device or manoeuvre being fully evaluated prior to widespread introduction. All the women would have been willing to be part of the clinical evaluation of the device, although there was concern around the timing of the explanation of this. There have now been several intrapartum clinical trials, which have used maternal assent with retrospective consent for interventions at critical times [40,41]; it could be assumed that consent for a study of devices and manoeuvres to facilitate delivery of the IFH could adopt a similar approach.

### 4.2. Strengths, Limitations, and Future Directions

This study was undertaken as part of participant engagement for a grant application. To our knowledge, this is the first study to use focus groups to capture the lived experiences of impacted foetal head for women and midwives involved. Other strengths include the facilitation of the group by a senior midwife and doctor (A.L.B., G.T.) who were able to probe deeply into the experiences for the midwives and the women.

Limitations include the relatively small sample size taken from one site; therefore, the results may only have local relevance. As such, generalisability of findings should be tested in other populations. Exclusion of medical and other allied health care professionals, as well as birth partners is also a limitation, which should be considered in future research.

Future research should focus on developing consensus and unified protocols for managing the IFH, together with training programmes for midwives, doctors and other health care professionals who encounter this emergency. All research in this area should include the voices of the women and their birth partners as well as multidisciplinary staff involved in caesarean births in late labour.

## 5. Conclusions

Our analysis details the knowledge gap for women regarding this common obstetric complication, and underlines a lack of consensus in the definition, management, and training for staff. Our study identifies knowledge gaps for women and their birth partners regarding IFH, describes the experience for midwives in this situation and outlines the importance of developing consensus around management of this common obstetric complication. It also confirms the need for alternative management strategies including new devices and the importance of these being fully evaluated prior to introduction into practice.

## Figures and Tables

**Table 1 ijerph-20-07009-t001:** Template analysis and data.

Core Principles of the Tydeman Tube	To Improve Outcomes for Mother and Baby in the Second Stage of Labour			To Reduce the Risk of Trauma to Mother and Baby in Complicated Births			To Increase Respectful Care for Women in Labour		
**Preliminary template**	Patient perspectives of complicated birth	Devices to deliver the foetal head	Manoeuvres to deliver an impacted foetal head	Expectation of trouble delivering the foetal head	Midwives’ role in management of impacted foetal head	Experience of impacted foetal head	Experience of ‘pushing up’	Debriefing after complicated birth	How partners coped
**Final template**	** *Current Knowledge of Impacted Foetal Head* **			** *Current Management of Impacted Foetal Head* **			** *Experiences and Outcomes of Impacted Foetal Head* **		
**Supporting data**	*From a patient’s perspective I think all they want is something that works………and is… you know, poses least risk to them or their baby…………. that any new thing needs properly investigating and researching to check how good it is before being introduced* (Woman 1) *My midwife was incredible though she stayed close by at all times and kept talking to me even when she needed to do other stuff……..her voice was really calming….. and of course [Doctor name] kept up a running commentary throughout [laughs]* (Woman 2) *I agree, I think any new thing that might make any part of labour and birth………. well, any bit of pregnancy safer and easier should be researched and once the evidence is available for doctors, midwives, and women to make proper choices, introduced, or not according to what is discovered* (Woman 4) *[Tydeman Tube]……….. I like the head and the fact it is flexible………….and does not look too big………. I think it looks more likely to work than the foetal pillow, you know……. just looking at it……… The handle is good too……I can see that even if I have to push up from below I will be outside the mum………still under the drapes possibly [laughs] but then I probably would be with the Foetal Pillow too [laughs]………..I’d like to see it work.* (Midwife 1) *…well anything that improved outcomes for mums and babies has got to be good………. I would want to see it [foetal pillow] first and again, maybe in mandatory training, have the chance to make sure I knew how to use it before being faced, you know, needing to use it, …. in the situation*. (Midwife 2)			*I was warned it could be tricky because of the forceps not working. ……But to be honest by that stage I just wanted it all to be over……………I’m not sure I understood what the doctor meant when he said that…* (Woman 5) *I am also aware that sometimes the team’s body language gives them away. You know, the anxious look in the eyes, the way people move, all those sort of things that partners, and women actually, really pick up on, but for staff are so very difficult to hide…………..I’m not sure how we deal with those, so I guess a device that might make the situation easier will help with that too.* (Midwife 1) *I don’t think any of us like being asked to do that………………..I was never trained how to do it…………..so I always,… always worry about the pressure on the baby’s head………I also have fairly short fingers so also worry about whether I am doing enough, …….or maybe anything at all* (Midwife 2) *I think we are also lucky here that we are a team that have you know worked together for years, we are all very much part of the community and as a team we support each other,……….I’m not sure that happens everywhere, so you know, if there is a difficult case, and you know in this situation babies have been damaged, and mothers too, I really feel that here we would all support the parents, obviously, but also support each other, you know get each other through it* (Midwife 2) *I think we need to go back to training both undergraduate and ongoing……………I’ve been qualified and working as a midwife for a while now, and I was certainly never taught how to push up from below, or indeed any management of the impacted foetal head……… I guess it must be covered in medical school, but having said that, many of the juniors do not seem to have been taught it, it comes as a shock when they actually have the problem in theatre. Maybe if there are going to be new devices one of the benefits of that will be, you know will be that difficult caesarean birth gets on the curriculum for both midwives and doctors* (Midwife 4)			*It was the midwife who told me ……….when she was checking the baby she said she was looking for bruising, which might come out later because they had had a bit if trouble getting the [babies name] head out……………..I just thought oh, OK* (Woman3) *But seriously,…I think he [partner] was quite traumatised by the experience overall………..as I said before [Dr name] was really calm throughout and I think he was reassured by that……….he didn’t realise the doc was probably really worried as he had this unexpected issue with a staff member [laughs]* (Woman 4) *yeah, yeah I think mine [partner] was freaked out in general by the time we got to theatre, and he was really upset when the baby couldn’t be born with the forceps…………I’m not sure we really discussed the actual, you know, the actual operation, the process at all* (Woman 5) *Yes…… my midwife did that too, she sort of disappeared under the drapes but I was not sure what was going on…………….good job the spinal was working [laughs]* (Woman 5) *From a staff point of view it is always difficult to gauge how the partner is doing at times like this, you know when you know there is a little complication and you are very aware that the partner is witnessing the whole thing, so you need, you know to give some explanation to him, but you don’t want to freak him out further…* (Midwife 1) *I think assisting at these sections…you know in when the head is wedged should be included in training………..not just basic training, but you know mandatory training…………it should be something you practice like shoulder dystocia and PPH…………..especially if you are not regularly working on the labour ward* (Midwife 4)		

## Data Availability

Data cannot be shared publicly because of potential identifiability. Data may be made available to researchers who meet the criteria for access to confidential data, upon reasonable request to the authors.

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
