# Peer review of "Experiences of Impacted Foetal Head: Findings from a Pragmatic Focus Group Study of Mothers and Midwives"

_ijerph, 2023, doi:10.3390/ijerph20217009_

Round 1

Reviewer 1 Report

Comments and Suggestions for Authors

Well, I was interested - but I have 2 -3 comments

1. The English is a bit too high for anyone who is not a primary English speaker - some areas can use simpler words. By the way, pragmatic was used so often - I was amused. 

2. I liked the fact that you stressed on the importance of awareness for patients and training for the midwives - this is the best take away from the paper. 

3. You intro said that the percentage of IFH  cesarean sections was about 9-11% but no where is there any talk about neonatal mortality, maternal complications. That will enhance the importance of knowing about this complication.

4. I would have like a little more description of the Tydeham tube and fetal pillow - in your paper - so that no one has doubts - or at least a reference to them. 

With a little modification - I would like to see this paper published

Comments on the Quality of English Language

very high flown English in the beginning - but got simpler in the results section onwards. 

Author Response

We have attached a point-by-point response.

Reviewer 2 Report

Comments and Suggestions for Authors

Thank you for the opportunity to review this interesting work. The study looked at the experience of women and midwives who had experienced a caesarean section complicated by IFH. As the authors argued, this is an understudied area in midwifery that deserves more attention due to its prevalence and lack of knowledge around it. I found the paper easy to read and all the methodological decisions such as the choice of a pragmatic approach, template analysis were all well justified. The conclusion aligns with the findings. Overall, this was a high quality paper. The only minor comment is the relevance and the decision to map the findings unto the Tydeman Tube. More information is needed on the Tydeman Tube and why it was used to orient the findings

Comments on the Quality of English Language

Some minor grammatical errors can be identified and eliminated by another proofreading. See below for example (Page 4, lines 167-168):

"A key feature of template analysis is that the preliminary template which is devised modified and augmented to ensure analytic completeness (King, 2012)".

Author Response

(The authors gave the same response as above.)

Reviewer 3 Report

Comments and Suggestions for Authors

Manuscript ID ijerph-2607919

It is a qualitative investigation that explored the experiences of the mother and midwives during cesarean delivery complicated by an impacted fetal head. Concluding that the midwives highlighted the lack of consensus regarding the definition, management and training, on the other hand the women anticipated that cesarean delivery at the last moment of delivery was simple and, therefore, they were unaware of this complication.

The manuscript requires clarification of some suggested observations.

In the abstract section

Whenever an abbreviation is used for the first time, it needs to be defined: IFH

In the introduction section

Whenever an abbreviation is used for the first time, it must be defined: CS

In the design section

What was the structured topic guide used in the focus group?

In the Ethical Considerations section

Other abbreviation used without previously defined (NHS)

What was the NHS approval number?

The characteristics of the place where the focus group activities were carried out must be specified.

Were both researchers moderators?

Moderators were either part of the community or external to the community.

Limitations

Why do you maintain that the sample size was small? This means that there was no saturation of the information provided by the participants.

It means that the purpose of understanding the historically constructed reality and analyzed in its particularities in light of the feelings and logic of its protagonists was not fulfilled.

Author Response

(The authors gave the same response as above.)

Round 2

Reviewer 1 Report

Comments and Suggestions for Authors

thank you for making it so much clearer now to understand. 

I think you have done a remarkable job with the editing. 

And I am happy you have acknowledged the small and perhaps biased sample. But I am also sure, there will be interest amongst readers to learn more about this technique.